# DATA OVERFITTING FOR ON-DEVICE SUPER-RESOLUTION WITH DYNAMIC ALGORITHM AND COMPILER CO-DESIGN

## ABSTRACT

Deep neural networks (DNNs) are frequently employed in a variety of computer vision applications. Nowadays, an emerging trend in the current video distribution system is to take the advantage of DNNs overfitting property to perform video resolution upscaling. By splitting videos into chunks and applying a super-resolution (SR) model to overfit each chunk, this scheme of SR models plus video chunks is able to replace traditional video transmission to enhance video quality and transmission efficiency. However, many models and chunks are needed to guarantee a high performance, which leads to tremendous overhead on model switching and memory footprints at the user end. To resolve such problems, we propose a Dynamic Deep neural network assisted by a Content-Aware data processing pipeline to reduce the model number down to one (Dy-DCA), which helps promote performance while conserving computational resources. Additionally, to achieve real acceleration on the user end, we design a framework that optimizes dynamic features (e.g., dynamic shapes, sizes, and control flow) in Dy-DCA to enable a series of compilation optimizations, including fused code generation, static execution planning, etc. By employ such techniques, our method achieves better PSNR and real-time performance (33 FPS) on an off-the-shelf mobile phone. Meanwhile, assisted by our compilation optimization, we achieve $1.7\times$ speedup while saving up to $1.61\times$ memory consumption.

## 1 INTRODUCTION

With the rapid advancement of artificial intelligence, Deep Neural Networks (DNNs) have emerged as a cornerstone technology in various computer vision tasks, revolutionizing the field of image processing. Among this, Video Super-Resolution (VSR) (Sajjadi et al., 2018; Tao et al., 2017; Caballero et al., 2017; Kim et al., 2018; Dai et al., 2017) has garnered increasing attention in recent years. Among the various approaches explored in the realm of VSR, a rising trend is focused on utilizing Super Resolution (SR) models to upscale the resolution of low-resolution (LR) videos instead of directly transmitting high-resolution (HR) videos (Yeo et al., 2018; Liu et al., 2021). This emerging representative aims to address the challenge of high bandwidth consumption between servers and clients, which often occurs when directly transmitting HR videos.

In the context of video transmission using neural networks, one prevalent approach is the utilization of conventional VSR models (Wang et al., 2019b;a), which are designed to cater to all types of videos. However, to achieve optimal performance, these models typically demand larger parameter sizes, rendering their deployment on mobile devices impractical (Wang et al., 2021; Hong et al., 2022; Zawad et al., 2022). Additionally, there is no assurance that a single model can consistently yield optimal results for all videos. To tackle these challenges, researchers have turned their attention to leveraging the overfitting property of DNNs. Instead of pursuing a one-size-fits-all approach, a novel strategy has emerged to utilize a dedicated model to overfit the whole video (Yeo et al., 2017; 2018). To enhance the quality of super-resolved video and reduce model size, raw videos are often split into segments based on time or spatial information (Li et al., 2022; Liu et al., 2021; Li et al., 2023), allowing each model to focus on a smaller segment. However, during the HR video recovery process, the frequent loading and unloading of numerous models can lead to significant overhead at the user end (Zawad et al., 2022). As shown in Figure 1, the overhead of switching model takes a

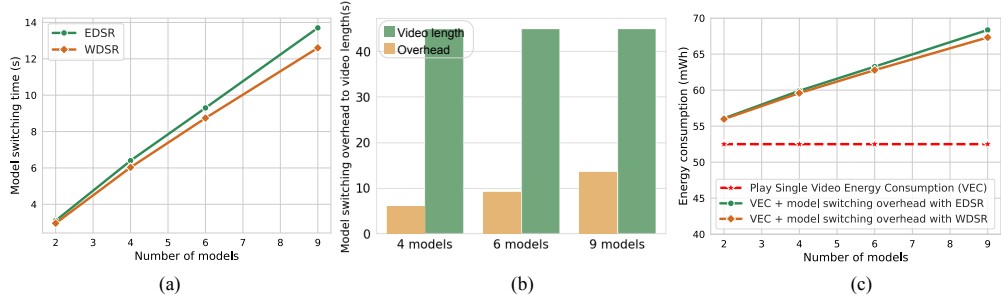

Figure 1: Model switching overhead on currently widely used backbones in video data overfitting. Figure (a) show the switching time in EDSR (Lim et al., 2017) and WDSR (Yu et al., 2018). Figure (b) demonstrates the comparison of video length and switching overhead. Figure (c) shows the total energy consumption brining by model switching.

large portion of total video length and brings more than 50% additional energy consumption, which is impossible to achieve a real-time performance as well as an efficient system.

Therefore, it is essential to find a video transmission framework capable of meeting the requirements for SR video quality and the resource demands of prevalent edge devices. Several key points are not fully discussed in previous works. (i) The number of models and corresponding model size should be minimized under certain SR video quality requirements. (ii) A corresponding algorithm-compiler-hardware optimization framework that can ensure a real-time system and reasonable on-device resource usage. To fulfill the above expectations, this paper proposes *Dy-DCA*, as shown in figure 2, which consists of a dynamic deep neural network and a fine-grained data preprocessing methodology to achieve better performance while minimizing the model switching overhead on the hardware side. Meanwhile, a complier-level optimization framework for accelerating dynamic DNNs is designed to achieve real-time inference and save memory consumption.

(i) **To minimize models while maintain high PSNR.** The prior art (Li et al., 2023) split video frames into evenly small patches and regroups them into different chunks according to texture complexity. Although this helps reduce the required model number and increases PSNR, these chunk & model pairs may still lead to I/O and model switching overhead at the user end (Zawad et al., 2022; Chan et al., 2022). Thus, in *Dy-DCA*, we propose a fine-grained data processing method that splits the frames into patches with uneven sizes (e.g., use large patches for monotonous backgrounds and smaller patches for detailed foregrounds), then overfits these data with a designed dynamic neural network, bringing the total transmitted model down to one. The uneven splitting minimizes the total number of patches, thereby reducing both server training effort and user-end I/O overhead. The dynamic neural network has a dynamic routing node and itself follows a tree structure to handle patches of different texture complexity.

(ii) **To ensure real-time performance on device.** Although dynamic DNN resolves the system overhead caused by model switching, the dynamic input shapes, and control flow in the model poses many challenges for the compiler-hardware-level optimizations (e.g., loop fusion (Zhu et al., 2021), execution order planning (Ahn et al., 2020), etc.). Due to very conservative assumptions and/or expensive analyses at runtime, current approaches (Abadi et al., 2016; Jiang et al., 2020b) face difficulty in achieving practical on-device efficiency. In this paper, we *finish the last piece of our design* for the system by proposing a nuanced approach that optimizes DNN dynamic features, which allows us to *close the loop* (algorithm, software, hardware) and achieving on-device intelligence. The foundation of our approach is an in-depth study of operators that form the basis for modern DNNs. These operators are classified into several groups on the basis of how the output shapes relate to input shapes and values. Under this classification, we introduce a data-flow analysis framework dedicated to inferring the shapes and dimensions of intermediate tensors. Subsequently, the outcomes of the analysis are used for enabling a number of compilation optimizations, which includes operator fusion and fused code generation, static execution planning, runtime memory allocation.

We summarize our contributions as follows :

- With the proposed context-aware data pipeline paired with dynamic DNN, patches with appropriate content and dimensions are directed to the suitable processing path, thus enhancing PSNR while reducing model shifting overhead.

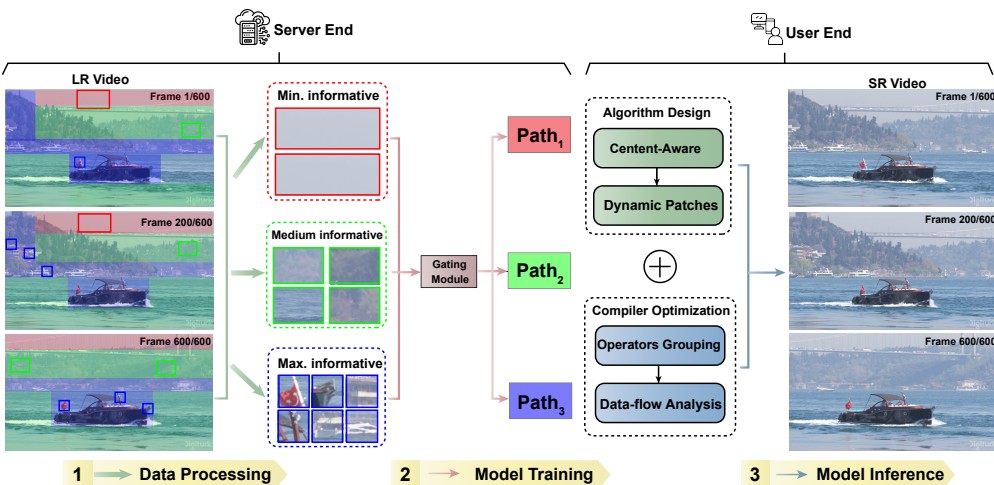

Figure 2: Overview of the proposed framework *Dy-DCA*. We split video frames into different shapes, and all patches will be distributed at a learnable gating module, then overfitted by a dynamic SR model. The dynamic SR model and LR patches will be delivered to users for video super-resolution. The on-device inference is accelerated by our designed compiler optimization framework.

- Given the existence of dynamic features, we introduce a data-flow analysis framework based on the classified DNN operators, which enables us to infer the shapes and dimensions of intermediate tensors. This design helps reduce the inference latency at the edge while reducing memory consumption.

- Based on the data-flow analysis framework, we implement a series of compiler-level optimizations (e.g. fused code generation, static execution planning, etc.) to solve the challenges brought by dynamic shape and routing, achieving $1.7\times$ overall speedup across various dynamic features.

## 2 ALGORITHM AND HARDWARE CO-DESIGN

### 2.1 MOTIVATIONS

The main promotions of previous arts concentrated on elevating the PSNR and expediting the training process (Li et al., 2022; Liu et al., 2021; Li et al., 2023; Yeo et al., 2018). An important aspect that hasn't received adequate attention is whether these improvements can actually be implemented on the devices used by end-users while maintaining the required quality. These methods often require multiple models for high performance and are sensitive to model number (Li et al., 2022; Liu et al., 2021; Yeo et al., 2018). Also, based on the result in Figure 1, numerous model will bring tremendous overhead, which may have a severe impact on the user end. Thus, our goal is to minimize the total number of models while maintaining good performance.

We investigate the patterns of these methods and find that there are two video segmentation methods used in their framework. Yeo et al. (2018) first operates on the temporal axis (e.g., every 5 seconds of content as a segment). As there may be consistent and repeated content in the whole video, different models may learn the same content at different segments, which is not effective. Also, a single frame covers information with large differences in texture complexity (e.g., foreground and background) (Bengio et al., 2007; Kumar et al., 2010; Fan et al., 2017; Toneva et al., 2018; Yuan et al., 2021), potentially making overfitting more difficult.

Li et al. (2023) splits video frames into evenly small patches (around $50\times50$) and regroups them into different chunks according to texture complexity. Although this method helps reduce the number of models, the small patches may carry similar information, which introduces meaningless computation and I/O overheads when training or inferring them. In order to reduce the need for multiple models, a better segmentation pattern and model structure need to be proposed.

## 2.2 Algorithm level optimization for hardware friendliness

To address the model switching overhead mentioned above, in this section, we propose a scalable dynamic deep neural network paired with a fine-grained data processing method that reduces the number of models down to one while maintaining good performance and a reasonable model size.

As shown in Figure 2, for an input video, we first split those frames into different shape patches. These patches contain various levels of texture complexity. (e.g., the minimum informative patches are concentrated in the background section, while the maximum ones are on the foreground object.) To achieve this, we propose a coarse- to fine-grained data processing pipeline to dynamically produce patches of different texture complexity. In our framework, frames will first be split into large patches, and evaluated by a general SR model to get the corresponding PSNR value for each patch. Guided by the PSNR values, we can roughly determine the texture complexity of each patch (Fan et al., 2017; Toneva et al., 2018; Yuan et al., 2021; Li et al., 2022; 2023). Specifically, for the patches where the PSNR value is greater than a certain threshold, as they contain less information, we will not further split those patches. For those lower ones, we will split them into smaller patches and follow the previous evaluation & split step. This forms an iterative processing pipeline to provide patches with different shapes and texture complexity. In this manner, patches with similar complexity features will almost all have the same shape. In *Dy-DCA*, each execution path is able to learn a similar distribution, which boosts the super-resolution performance. Also, the total number of patches in a video drops a lot compared to the method (Li et al., 2023), thus reducing the I/O pressure on the user end.

To overfit the above patches with different shapes, we propose a dynamic deep neural network with different paths and a routing node. Specifically, with multiple paths for various resolutions and a learnable routing node to distribute input patches to corresponding paths, our proposed *Dy-DCA* resolves the system overhead caused by model switching. Meanwhile, taking advantage of modular SR neural network design (Lim et al., 2017; Yu et al., 2018; Zhang et al., 2018), we delicately design the structure of each path to maintain better performance and a reasonable model size for the resource-constrained devices.

## 2.3 Compiler level optimization to better support algorithm

In section 2.2, we utilize a dynamic neural network to resolve the system overhead caused by model switching. However, we have different shape patches as input (dynamic input), and these patches will be distributed to different paths (routing), introducing uncertainty into model execution. Thus, a compiler-level optimization method should be proposed to resolve the challenges brought by dynamic input and routing.

### 2.3.1 DNN Operator Classification

In dynamic DNNs, each tensor can be categorized into input tensor ($I$) and output tensor ($O$) at operator ($T^l$), where $l$ is the operator index. We denote the shape of an input tensor as $I(S)$ and its corresponding value as $I(V)$. Similarly, for an output tensor, we have $O(S)$ and $O(V)$ to represent shape and value, respectively.

Our key observation is that dynamism (routing, input) brings uncertainty to optimization. For instance, loop fusion (Zhu et al., 2021) relies on knowledge of the identical index space between two loops, typically corresponding to the dimensions of respective input tensors. Likewise, planning execution order (Ahn et al., 2020) to minimize memory usage or organizing memory allocation (Pisarchyk & Lee, 2020) is hindered in the absence of static knowledge regarding tensor sizes.

Thus, in order to help predict intermediate tensor shape and value, we first group DNN operators into four types: Input Shape Determined Output ($I(S) \rightarrow O(S, V)$), Input Shape Determined Output Shape ($I(S) \rightarrow O(S)$), Input Shape & Value Determined Output Shape ($I(S, V) \rightarrow O(S)$), and Execution Determined Output ($Exec \rightarrow O(S, V)$).

**(i) Input Shape Determined Output:** The output tensor shapes are determined by the input tensor shape, but the input values do not impact the output. The representative operators include `Shape` and `Eyelike`.

Table 1: DNN Operators Classification. These operators are from ONNX (Open Neural Network Exchange) (ONNX).

| Operator Class | Example of Operators | Representatives |
|---|---|---|
| I(S) → O(S, V) | Shape, ConstantOfShape, Eyelike | Shape |
| I(S) → O(S) | Add, AveragePool, Cast, Concat, Conv, Elementwise w/ broadcast, Gather, MatMul, MaxPool, Reduce, Relu, Round, Sigmoid, Softmax | Conv, MatMul |
| I(S,V) → O(S) | Expand, GroupNormalization, MaxUnpool, Onehot, Range, Reshape, Resize, Slice, TopK, Upsample | Reshape, Range |
| Exec → O(S, V) | If, Loop, NMS, Nonzero, <Switch, Combine> | If, Loop |

**(ii) Input Shape Determined Output Shape:** The output shapes are dependent on the input shapes, much like in the preceding category. The output values, however, depend on all the input values. Examples include `Conv`, `Add`, and `Pooling`. The significance of this category, as compared to the next set of categories, is that if the input shape of this operator is known, compiler optimizations (e.g., operator fusion, execution/memory optimizations) are enabled.

**(iii) Input Shape & Value Determined Output Shape:** The output values are dependent on the input shapes and all the input values, just like the previous category. The distinction is that only a portion of the input data is used by the output shapes. Examples include `Extend` and `Range`.

**(iv) Execution Determined Output:** Similar to the previous two categories, the output values rely on the input shapes and all the input values. Examples include `Nonzero` and `If`.

### 2.3.2 DATA-FLOW ANALYSIS

In this section, we present a novel data-flow analysis methodology to infer the intermediate result tensor shape based on the DNN operator classification discussed above. Such an analysis enables a number of optimizations, including dynamic DNN operator fusion, execution path planning, etc.

Our main finding is that, for many operators and operator combinations (such as an input shape determined output operator and an input shape determined output shape operator), it is possible to infer the shape of the intermediate result tensor to some extent even without knowing the input tensor shape.

**Data-flow domain:** To infer the shape and value of the intermediate tensor, we first define the value domain of tensor shape (S) and tensor value (V) for subsequent optimization algorithms. As shown in Figure 3, the lattice includes known constants, symbolic constants, and operator-inferred constant. The lattice also includes undefined and not-a-constant (nac) at the top and bottom, respectively.

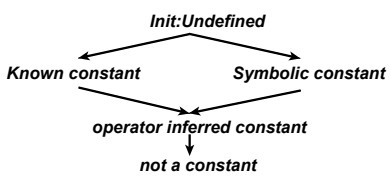

Figure 3: The lattice of the data-flow domain.

**Transfer function:** The transfer function is used to transfer the Shape and Value from the input tensor to the output tensor based on the operator type. In our proposed dynamic neural network, there are two kinds of transfer functions: `Update` and `Merge`. `Update` transfers from the input tensor to the output tensor for an individual operator. `Merge` operates on branch control flow and merges (output) tensors from multiple possible execution paths.

**Algorithm for data-flow analysis:** The computation graph (G) of a dynamic neural network can be viewed as a Directed Acyclic Graph (DGA). ① We first sort the operators in G in depth-first order and initialize the output shape and value map of each operator as undefined. ② If the operator is a control-flow node (like Combine or Switch), it needs to call the Merge function to merge analysis results from multiple control-flow paths. ③ For non-control-flow operators, we send them to forward transfer functions. These functions are based on dynamism classification of DNN operators in Table 1. For example, if the operator in the case "Input Shape Determined Output Shape" like Conv and Add, the function will return the predicted shape. ④ Then, for the case of "Input Shape Determined Output", if the value domain of operator shape is not undefined or nac, we then set a

symbolic value to the output tensor value to facilitate subsequent analysis. The algorithm will iteratively process ②-④ until there are no updates on the shape and value of the output tensor of an operator.

### 2.3.3 OPERATOR FUSION

A common challenge encountered in dynamic DNNs are the absence of knowledge regarding the tensor shapes of two operators. In such cases, the DNN compiler faces limitations, as it is either unable to perform fusion between these operators or has to generate a multitude of code versions, each accommodating a possible combination of shapes for the two operators. In reality, when we are dealing with the merging of more than two operators, the need to generate separate code for all the potential combinations becomes quite extensive.

This problem can be resolved by our suggested data-flow analysis, which makes use of (potentially symbolic) shape information. Fusion can be enabled and/or made simpler by information like the fact that the two operators have tensors of the same shape, even if the precise dimensions are not known until runtime.

### 2.3.4 STATIC EXECUTION PLANNING

The computational graph of dynamic DNN supports a variety of orderings for the execution of operators. The ordering decision affects the peak memory usage, which further affects the effectiveness of the cache and the execution latency. However, it has been proved that finding an optimal path in the computational graph is a NP-hard problem (Ahn et al., 2020). Therefore, it is hard to find an optimal plan for modern large DNNs with more than hundreds of operators.

In our data-flow analysis, the general assumption is that a method based on graph partitioning is appropriate because a globally optimal solution is impractical. It turns out that the analysis results can direct the segmentation of the graph, as well as the choice of solution within each sub-graph. The main reason lies in the value domain (undefined, known constant, symbolic constants, operator-inferred constants, and not a constant) we define before, which facilitates the generation of an optimal execution plan. For example, in a sub-graph $G\_sub$, if all tensors in $G\_sub$ are known, the optimal execution plan for $G\_sub$ can be obtained statically by an exhaustive search. If there exist known constant, symbolic constants, and operator-inferred constants, it is still possible to generate a close optimal execution under certain requirement like memory usage and latency limitation. For those operators have a not constant output tensor shape, it provides an opportunity to partition the original graph into sub-graphs that can be independently analyzed.

## 3 EXPERIMENTAL RESULTS

In this section, we evaluate Dy-DCA on multiple videos to show its superiority. Also, to demonstrate the effectiveness of our proposed algorithm and compiler co-design approach, we deploy Dy-DCA on an off-the-shelf mobile phone. The detail information of dataset and implementation can be found in Section 3.1. We compare our method with current multimodel method (Li et al., 2023; Yeo et al., 2018; Li et al., 2022), and the main results are shown in Section 3.2. In Section 3.3, we deploy our model on edge and achieve real-time inference speed, which shows the advantages of our proposed compiler optimization.

### 3.1 EXPERIMENT SETTINGS

To evaluate the overfitting performance of our framework, we we adopt the UVG (Mercat et al., 2020) and VSD4K (Liu et al., 2021) dataset. UVG dataset contains 16 test video sequences and each video has multiple resolution and bitrate to choose. In VSD4K video dataset, there are 6 video categories and each of the category contains various video lengths. We set the resolution for HR videos to 1080p and bitrate at 10bit, and LR videos are generated by bi-cubic interpolation to match different scaling factors.

We utilize EDSR (Lim et al., 2017), and WDSR (Yu et al., 2018) as our backbone. Both of them use a modular design, which provides flexibility in our design of dynamic neural networks. During training, we split frames into different size of patches. The PSNR threshold we set for splitting is 40

Table 2: Comparison results of Dy-DCA with different data overfitting methods.

| Model | Data Scale | UVG-Beauty | | | UVG-Bosphorus | | | UVG-HoneyBee | | |
|---|---|---|---|---|---|---|---|---|---|---|
| | | ×2 | ×3 | ×4 | ×2 | ×3 | ×4 | ×2 | ×3 | ×4 |
| EDSR | NAS 43 | 45.74 | 42.71 | 40.52 | 44.99 | 41.47 | 38.04 | 44.54 | 41.51 | 40.23 |
| | EMT 25 | 46.84 | 43.79 | 41.80 | 46.12 | 42.33 | 39.14 | 45.21 | 42.29 | 39.38 |
| | STDO 24 | 47.02 | 44.12 | 42.03 | 46.64 | 42.72 | 39.71 | 45.82 | 42.74 | 42.15 |
| | **Ours** | **47.21** | **44.42** | **42.35** | **46.87** | **43.06** | **40.25** | **46.02** | **43.24** | **42.74** |
| WDSR | NAS 43 | 46.23 | 43.24 | 41.10 | 45.45 | 42.01 | 38.92 | 44.89 | 41.97 | 41.08 |
| | EMT 25 | 47.04 | 44.17 | 42.40 | 46.42 | 42.86 | 39.92 | 45.87 | 43.01 | 42.12 |
| | STDO 24 | 47.50 | 44.73 | 42.88 | 46.98 | 43.06 | 40.22 | 46.10 | 43.29 | 42.74 |
| | **Ours** | **47.64** | **44.85** | **43.10** | **47.10** | **43.31** | **40.29** | **46.33** | **43.52** | **43.02** |
| | | game-45s | | | inter-45s | | | vlog-45s | | |
| | | ×2 | ×3 | ×4 | ×2 | ×3 | ×4 | ×2 | ×3 | ×4 |
| EDSR | NAS 43 | 43.22 | 36.72 | 34.32 | 43.31 | 35.80 | 32.67 | 48.48 | 44.12 | 42.12 |
| | EMT 25 | 44.04 | 37.89 | 35.27 | 43.89 | 36.21 | 33.07 | 48.86 | 44.71 | 42.80 |
| | STDO24 | 45.65 | 39.93 | 37.24 | 44.52 | 38.28 | 35.51 | 49.84 | 45.47 | 43.07 |
| | **Ours** | **45.72** | **40.17** | **37.60** | **44.74** | **38.53** | **35.77** | **49.89** | **45.61** | **43.19** |
| WDSR | NAS 43 | 43.70 | 37.25 | 34.93 | 43.41 | 36.05 | 33.11 | 48.52 | 44.75 | 42.80 |
| | EMT 43 | 44.47 | 38.14 | 35.72 | 43.92 | 36.73 | 33.47 | 49.07 | 44.72 | 42.87 |
| | STDO24 | 45.71 | 40.33 | 37.76 | 44.54 | 38.72 | 36.03 | 49.76 | 45.95 | 43.99 |
| | **Ours** | **45.86** | **40.51** | **37.98** | **44.76** | **39.02** | **36.44** | **49.84** | **46.13** | **44.12** |

and 30. Specifically, patches with PSNR larger than 40 will not be split. The next round larger than 30 will not be split. (Nasution et al., 2018) Regarding the hyperparameter configuration of training the SR models, we follow the setting of (Lim et al., 2017; Yu et al., 2018; Liu et al., 2021). We adopt Adam optimizer with $\beta_1 = 0.9$, $\beta_2 = 0.009$, $\epsilon = 10^{-8}$ and we use L1 loss as loss function.

## 3.2 EVALUATION ON VSD4K AND UVG DATASETS

In this part, we sample three video categories from VSD4K and UVG and tested them on 45-second videos (VSD4K) and 20-second videos (UVG), respectively. Our results are shown in Table 2. The visual results are shown in Figure 4. We compare with the state-of-the-art DNN-based SR video overfitting methods, such as NAS (Yeo et al., 2018) that splits a video into multiple chunks in advance and overfit each of each chunk with independent SR model, EMT (Li et al., 2022) utilizes same training pattern while take advantage of meta learning to accelerate training, and STDO (Li et al., 2023) that uses texture information to divided video chunk and overfits with multiple models. As we can see, we achieve higher PSNR across these different videos.

## 3.3 DEPLOYMENT ON MOBILE DEVICES

The on-device evaluations are conducted using a OnePlus 11 mobile phone, equipped with a Snapdragon 8 Gen 2 processor (2023.). This processor is built on one Cortex-X3 based prime core, four Cortex A-715 and A-710 based performance core, three Cortex A-510 based efficient core, and paired with a Qualcomm Adreno 740 GPU, delivers superior performance while maintaining efficient power consumption. We tested our model on Alibaba MNN (Jiang et al., 2020a) with standard configuration, which utilizing 4 threads and employing FP16 precision to optimize computational efficiency. The inference process is executed 20 times, with the first 5 iterations serving as a warm-up phase to ensure system stability and to negate the effects of start-up transients on the experimental outcomes. The results are then averaged to account for any variations, and measured the performance under the designated conditions. The main points of our assessment are to methodically analyze model shift overhead, run-time memory usage, power consumption, and image loading overhead.

**Memory and latency result.** In Table 3, the memory consumption and end-to-end execution latency are tested by MNN. As *Dy-DCA* has different input patches shape and these patches will be sent to different paths, we randomly chose 50 frames to inference then averaged them. Since NAS, EMT and STDO utilize static DNN, they do not support dynamic input shapes and routing, thus we do not show the results of these methods on our proposed optimization framework. Overall, with our designed compiler optimization framework, we achieve 1.7× speedup and 1.61× memory reduction compared with MNN. The average inference speed on mobile GPU is 30ms, which achieves real-time requirement (Zhan et al., 2021).

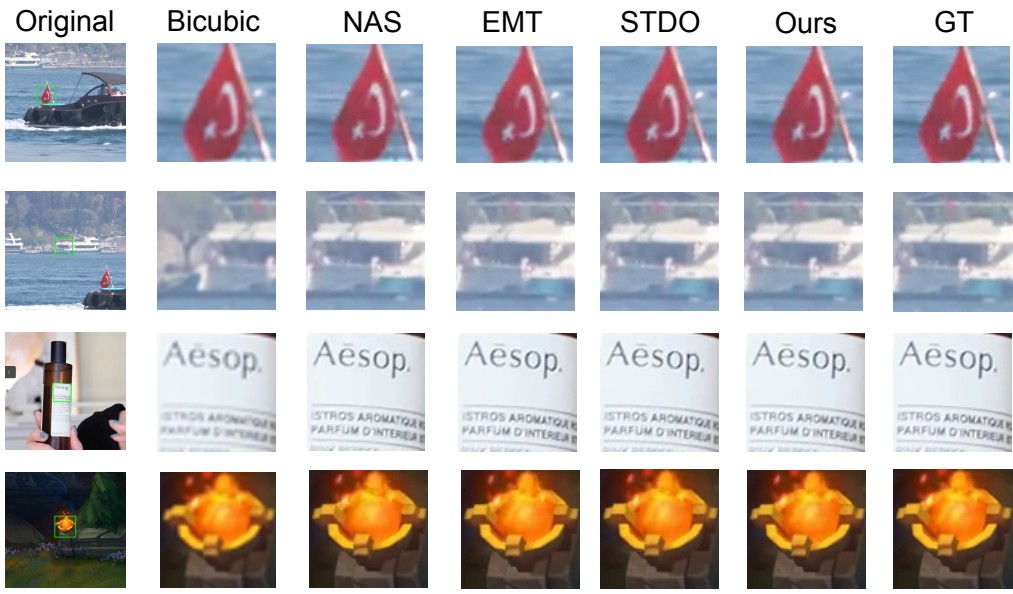

Figure 4: Super-resolution quality comparison on Dy-DCA and baseline methods.

Table 3: Memory consumption and end-to-end execution latency comparison among MNN and our proposed data-flow analysis framework. S stands for the number of parameter of a 16-4 WDSR (Yu et al., 2018) model. We test on the 45 seconds video. The "/" format is mobile CPU/GPU.

| Method | Model Size | MNN(MB) | | Ours(MB) | | MNN(ms) | | Ours(ms) | |
|---|---|---|---|---|---|---|---|---|---|
| | | Min | Max | Min | Max | Min | Max | Min | Max |
| NAS[43] | 9S | 79/415 | 79/415 | - | - | 457/271 | 476/275 | - | - |
| EMT [25] | 9S | 79/415 | 79/415 | - | - | 457/271 | 476/275 | - | - |
| STDO [24] | 4S | 72/175 | 72/175 | - | - | 334/158 | 342/162 | - | - |
| Ours | S | 66/170 | 69/184 | 47/103 | 53/116 | 66/48 | 82/54 | 52/28 | 59/32 |
| Our's framework speedup | | 1.45× - 1.61× | | 1 | | 1.30× - 1.70× | | 1 | |

**Overhead analysis.** In Table 4, As our proposed *Dy-DCA* reduced the model down to one and provide a fine-grained video frame splitting, we analyze the overhead in terms of model switching and I/O. As STDO also split video frames into patches, compare with it, we achieve 4× saving on model switching overhead and 7× saving on I/O.

Table 4: The comparison of model switching and I/O overhead between STDO and our method.

| Method | Model Number | switching Overhead | Patches Number | Average I/O Overhead | PSNR |
|---|---|---|---|---|---|
| STDO [24] | 4 | 4× | 1.2×10e5 | 6× | 47.50 |
| Ours | 1 | 1× | 2.16×10e4 | 1× | 47.64 |

## 4 RELATED WORKS

### 4.1 VIDEO SUPER RESOLUTION (VSR)

The VSR methods mainly drive from Single Image Super Resolution (SISR) framework (Liu et al., 2022). Some of the works described above that were primarily created for SISR, including EDSR (Lim et al., 2017) and WDSR (Yu et al., 2018), all have results on VSR. Several of the recent VSR works perform alignment to calculate optical flow by DNNs in order to estimate the motions between images (Sajjadi et al., 2018; Tao et al., 2017; Caballero et al., 2017; Kim et al., 2018).

However, accurate optical flow may not be easy to compute for videos with occlusion and large motions. Another method to perform alignment is called deformable convolution methods, which is first proposed by Dai et al. (2017). The Deformable convolution (DConv) (Dai et al., 2017) was first used to deal with geometric transformation in vision tasks because the sampling operation of CNN is fixed. TDAN (Tian et al., 2020) applies DConv to align the input frames at the feature level, which avoids the two-stage process used in previous optical flow based methods. However, these models incorporate with DConv may suffer from high computation complexity and difficulty for convergence. To increase the robustness of alignment and account for the visual informativeness of each frame, EDVR (Wang et al., 2019b) uses their proposed Pyramid, Cascading and Deformable convolutions (PCD) alignment module and the temporal-spatial attention (TSA) fusion module.

## 4.2 DEVELOPMENT OF DYNAMIC DNN

The development of dynamic DNNs is extensive, including recurrent neural networks (RNNs) and their derivatives, along with instance-specific dynamic models, spatial-oriented dynamic networks, and time-oriented dynamic models (Han et al., 2022). Our motivation for utilizing DNNs in super-resolution stems from their adaptability and the comprehensive range of dynamic capabilities they offer, allowing for enhanced performance in reconstructing high-resolution details from low-resolution inputs. This survey (Han et al., 2022) highlights numerous studies pertinent to dynamic inference, illustrating the establishment of model aggregations through either cascaded or concurrent configurations and the selective activation of models based on input conditions. Additionally, this advancement underscores the significance of deploying Spiking Neural Networks (SNNs), which conduct data-driven inference through the propagation of pulse signals (Ghosh-Dastidar & Adeli, 2009). This review also underscores several pivotal publications in these domains, such as Dynamic Network Surgery (Guo et al., 2016), Spatially Adaptive Computation Time (SACT) (Figurnov et al., 2017), Dynamic Conditional Networks (DCNs) (Zhao et al., 2018), Dynamic Filter Networks (Zhou et al., 2021), Dynamic Convolutional Neural Networks (DCNNs) (Chen et al., 2020b), Dynamic Routing Between Capsules (Sabour et al., 2017), Dynamic Skip Connections (DSCs) (Gui et al., 2018), and Dynamic Time Warping Networks (DTWNs) (Cai et al., 2019). The discussion concludes by reflecting on the unresolved challenges in this field and suggesting intriguing paths for future research. These include the development of theories for dynamic networks, the crafting of efficient decision-making, and the diversified application exploration across various disciplines.

## 4.3 CONTENT-AWARE DNN

It is not possible to develop a DNN model that can efficiently handle all web video. To ensure reliability and performance, Yeo et al. (2018) suggests that the video delivery system take into account employing DNN models to overfit each video chunk. Several livestreaming and video streaming applications (Yeo et al., 2020; Xiao et al., 2019; Kim et al., 2020; Chen et al., 2020a; Dasari et al., 2020) make use of overfitting property to guarantee great client performance. Kim et al. (2020) proposes a live video ingest framework, which adds an online learning module to the original NAS (Yeo et al., 2018) framework to further ensure quality. NEMO (Yeo et al., 2020) selects key frames to apply super-resolution. This greatly reduces the amount of computation on the client sides. CaFM (Liu et al., 2021) splits a long video into several time-based chunks and design a handcrafted layer along with a joint training technique to reduce the number of SR models and improve performance. EMT (Li et al., 2022) proposes to leverage meta-tuning and challenge patches sampling technique to further reduce model size and computation cost. STDO (Li et al., 2023) takes spatial information as well as temporal information into account to further enhance model performance.

## 5 CONCLUSION

In this paper, we introduce a content-ware dynamic DNN to overfit videos. This design reduces the required model number down to one, thus reducing the model switching overhead at the user end. In order to resolve the challenges brought by dynamic input patches and routing in dynamic DNN, we propose a data-flow analysis framework to predict the shape and value of intermediate tensor. Subsequently, the outcomes of the analysis are used to enable a number of compilation optimizations, which achieve real-time performance on the edge.

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
