# OpenReview forum: "Data Overfitting for On-Device Super-Resolution with Dynamic Algorithm and Compiler Co-Design"
_ICLR.cc/2024/Conference — ICLR 2024 Conference Withdrawn Submission_

### Official Review · Reviewer_PF2v · 2023-11-01

**Soundness:** 3 good
**Presentation:** 3 good
**Contribution:** 2 fair
**Rating:** 6
**Confidence:** 3

**Summary:**

The paper introduces a novel framework, Dy-DCA, which combines a Dynamic Deep Neural Network with a Content-Aware data processing pipeline to enhance on-device super-resolution for videos. This approach aims to address the challenges of model switching overhead and memory footprint associated with traditional video super-resolution methods that rely on splitting videos into chunks and overfitting a super-resolution model to each chunk.

**Key Contributions:**
- Dynamic Neural Network with Content-Aware Data Processing: The paper proposes a scalable dynamic deep neural network paired with a fine-grained data processing method, significantly reducing the number of required models while maintaining high performance and reasonable model size. This is achieved by dynamically producing patches of different texture complexity and overfitting these patches with a designed dynamic neural network.

- Compiler-Level Optimization: To accommodate the dynamic nature of the proposed neural network and to ensure real-time performance on devices, the paper introduces a compiler-level optimization framework. This framework optimizes dynamic features such as input shapes and control flow, enabling a series of compilation optimizations that result in faster execution and reduced memory consumption.

- Enhanced Video Quality and Efficiency: By employing the proposed framework and optimizations, the paper claims to achieve better PSNR and real-time performance on mobile devices, along with significant improvements in speed (1.7× overall speedup) and memory consumption (up to 1.61× savings).

**Strengths:**

**Originality:**
The paper introduces a novel framework, Dy-DCA, which combines a dynamic deep neural network with a content-aware data processing pipeline for on-device super-resolution. This approach is original as it addresses the common issue of model switching overhead in video super-resolution, reducing the number of required models to one. The integration of a compiler-level optimization framework to support the dynamic nature of the neural network adds a unique dimension to the work, showcasing an innovative solution to a well-known problem in video super-resolution.

**Quality:**
The paper appears to be of high quality, providing a comprehensive and well-thought-out solution to enhance video quality and efficiency in transmission. The proposed Dy-DCA framework and the associated compiler-level optimizations are grounded in solid theoretical and practical considerations, with claims of significant improvements in PSNR, real-time performance on mobile devices, and resource efficiency.

**Clarity:**
The paper is well-structured and articulates the problem, proposed solution, and contributions clearly. The use of figures and step-by-step explanations aid in understanding the complex concepts involved in the Dy-DCA framework and compiler-level optimizations. However, the depth of the content may require readers to have a substantial background in deep learning, computer vision, and compiler optimizations.

**Significance:**
The paper holds significant potential to impact the field of video streaming and super-resolution, addressing critical issues of video quality and transmission efficiency on edge devices. By reducing model switching overhead and memory footprint, the paper presents a solution that could lead to more efficient and effective video super-resolution applications, especially in real-time scenarios on resource-constrained devices.

**Weaknesses:**

- It would be more comprehensive to add ablation studies to understand the contribution of each component of the Dy-DCA framework and the compiler-level optimizations would offer a clearer picture of their individual and combined effects on performance and quality.

- The paper could benefit from experiments to evaluate the scalability and efficiency of the proposed framework on a variety of hardware architectures, including both high-end and low-end devices, would provide a comprehensive view of its applicability and performance across different scenarios.

- The PSNR gain is marginal, though the speedup is shiny.

**Questions:**

Could you clarify how the PSNR gain in table 2 and 4 are *significant*? From Figure 4, visually the performance seems close to other approaches.

---

> ### Author Response · Authors · 2023-11-18
>
> Dear Reviewer PF2v,
>
> Thank you for your review and suggestions on our paper.
>
> **W1:It would be more comprehensive to add ablation studies to understand the contribution of each component of the Dy-DCA framework and the compiler-level optimizations would offer a clearer picture of their individual and combined effects on performance and quality.**
>
> Thank you for the suggestion. The high video quality of Dy-DCA attributes to the connection of dynamism of the neural network to the context of video content. The most important factors that affects the video quality would be the the granularity of how we group different content to groups and feed them to different dynamic neural network paths. We analyze the different number of paths in the network structure of Dy-DCA. As shown in the following table, more paths bring higher PSNR by introducing fine-grained data patch groups with more trainable parameters, which sacrifices some latency at the user end. Due to limited time, the results in our submitted paper were mainly conducted on a two-path scenario, but we will add the ablation study in the final revision.
>
> | Method | Path Number | PSNR |  &ensp;Ours(ms) |
> | :--- | :---: | :---: | :---: |
> |  | |  | Min &ensp; Max |
> | Dy-DCA(WDSR) | 2 | 45.86 | 28 &ensp; 32 |
> | Dy-DCA(WDSR) | 3 | 46.33 | 33 &ensp; 37 |
> | Dy-DCA(WDSR) | 4 | 46.51 | 37 &ensp; 42 |
>
> **W2:variety of hardware architectures, including both high-end and low-end devices**
>
> Thank you for the suggestion. Our design mainly targets mobile device. In order to demonstrate high-end and low-end difference, we chose a Samsung Galaxy S10 cell phone with Qualcomm Snapdragon 855 mobile platform[R1] released 5 years ago, which can be viewed as the low-end devices in current stage. The high-end hardware device is OnePlus 11 mobile phone equipped with a Snapdragon 8 Gen 2 processor [R2] used in our paper. As shown in the following table, our method is able to achieve 1.35$\times$ speedup on Snapdragon 855 and 1.32$\times$ speedup on Snapdragon 8 Gen 2. Thus, the compiler-level optimization design is able to accelerate inference speed among devices with different hardware conditions.
>
> | Method |  Device Processor |   MNN(ms) |  Ours(ms) |
> | :---: | :---: | :---: | :---: |
> | |  |Min &ensp; Max | Min &ensp; Max |
> | Dy-DCA | Snapdragon 855 | 127 &ensp; 138 | 93 &ensp; 104 |
> | Dy-DCA | Snapdragon 8 Gen 2 | 66 &ensp; 82 | 52 &ensp; 59 |
>
> [R1]Snapdragon 855. https://www.qualcomm.com/products/mobile/snapdragon/smartphones/snapdragon-8-series-mobile-platforms/snapdragon-855-mobile-platform
>
> [R2]Snapdragon 8 gen 2. https://www.qualcomm.com/products/mobile/snapdragon/smartphones/snapdragon-8-series-mobile-platforms/snapdragon-8-gen-2-mobile-platform.
>
> **W3: The PSNR gain is marginal, though the speedup is shiny.**
>
> Our design seeks to achieve speedup and the real-time inference on the mobile devices. In our initial experimental settings, we aim to achieve highest acceleration ratio. So we set the number of paths to 2 in the Dy-DCA network structure, which is coarse-grained for video content clustering but achieves highest acceleration speed since the computation graph is less complex. We further perform more experiments, and we find that more paths can improve the PSNR significantly while only increasing latency slightly. The reason is that more fine-grained data patch groups with more trainable parameters can improve learning performance, and more paths will not affect the execution efficiency since we still use one single model for inference. Although other method achieve similar PSNR in Table 2, our on-device latency is roughly 2 to 5 times faster, as shown in Table 3. Also, other method are far from real-time requirement.
>
> **Q1: visually the performance seems close**
>
> The visual results come from the $\times2$ scaling factor. Therefore, the PSNR is very high (the PSNR of most video topics is more than 45dB) and the performance distance among different methods is narrow. Another potential reason is that we randomly pick the frames in a video to do a visual comparison, while the PSNR results are averaged over each frame in the video. We will consider using the $\times4$ scaling factor for a more pronounced comparison.

---

> ### Author Response · Authors · 2023-11-20
>
> Dear Reviewer PF2v,
>
> We highly value the time and dedication you've invested in reviewing our submission. As the deadline for the ICLR rebuttal nears on the 22nd, your prompt feedback would immensely aid us in writing a thorough response, allowing us to comprehensively address the key points you have highlighted. Thanks.

---

### Official Review · Reviewer_nwgm · 2023-11-03

**Soundness:** 2 fair
**Presentation:** 3 good
**Contribution:** 2 fair
**Rating:** 5
**Confidence:** 4

**Summary:**

The paper proposes a Dynamic Deep neural network assisted by a Content-Aware data processing pipeline to reduce the number of models down to one (Dy-DCA), while still maintaining good performance. Meanwhile, the paper designs a framework that optimizes dynamic features (e.g., dynamic shapes, sizes, and control flow) in Dy-DCA to enable a series of compilation optimizations, including fused code generation, static execution planning, etc. The performance of the proposed solution is evaluated on two datasets.

**Strengths:**

1. The proposed solution achieves better PSNR and real-time performance compared to traditional video transmission, while reducing the number of models needed for high performance down to one and conserving computational resources.

2. The proposed solution optimizes dynamic features (such as dynamic shapes, sizes, and control flow) to enable a series of compilation optimizations (including fused code generation, static execution planning, etc.), which helps achieve acceleration on the user end.

**Weaknesses:**

1. The paper lacks a detailed discussion of the limitations and potential drawbacks of the proposed solution.

2. The paper does not provide a detailed comparison of the proposed solution with other state-of-the-art methods in terms of model size and computational complexity.

3. The paper only provide PSNR performance. Additional subjective quality (such as MS-SSIM) should be measured.

4. The paper does not provide ablation experiments.

**Questions:**

Please refer to weaknesses.

---

> ### Author Response · Authors · 2023-11-18
>
> Dear Reviewer nwgm,
>
> We sincerely appreciate your thoughtful comments about our work. Regarding the questions you raised, we believe they are important points that merit further attention.
>
> **W1: The paper lacks a detailed discussion of the limitations and potential drawbacks of the proposed solution.**
>
> Thanks for the suggestion. In our paper, we only demonstrate the compiler acceleration framework on video resolution tasks based on CNN. As the vision transformer performs well on many visual tasks, the transformer-based dynamism and video super-resolution need further discussion.
>
>
> **W2:The paper does not provide a detailed comparison of the proposed solution with other state-of-the-art methods in terms of model size and computational complexity.**
>
> Thank you for your suggestion. The model we choose is based on the previous works by STDO[R1], EMT[R2] and CaFM[R3], and we compare with STDO and EMT these two state-of-the-art methods with the same model architecture.
> In terms of model size, we do show the model size in Table 3 and Table 4. In Table 3, we achieve lower memory consumption and end-to-end latency with a smaller model size. In Table 4, we achieve higher PSNR and lower I/O overhead with a smaller model size. As for computational complexity, we believe that it is more convincing to show an on-device end-to-end inference speed (as shown in Table 3) as a stronger demonstration beyond pure theoretical computational complexity.
>
> [R1]Li G, Ji J, Qin M, et al. Towards High-Quality and Efficient Video Super-Resolution via Spatial-Temporal Data Overfitting, CVPR 2023.
>
> [R2]Li X, Liu J, Wang S, et al. Efficient meta-tuning for content-aware neural video delivery, ECCV 2022.
>
> [R3]Liu J, Lu M, Chen K, et al. Overfitting the data: Compact neural video delivery via content-aware feature modulation, ICCV 2021.
>
>
> **W3:The paper only provide PSNR performance. Additional subjective quality (such as MS-SSIM) should be measured.**
>
> Thank you for the suggestion. We list partial results on the UVG dataset with the scaling factor $\times2$ with WDSR backbone. We will add full SSIM results in the later revision.
>
> | Method | UVG-Beauty   | UVG-Bosphorus  | UVG-HoneyBee |
> | :--- | :--- | :--- | :--- |
> | NAS | 46.23 / 0.9873 | 45.45 / 0.9856 | 44.89 / 0.9851 |
> | EMT | 47.04 / 0.9887 | 46.42 / 0.9879 | 45.87 / 0.9870 |
> | STDO | 47.50 / 0.9890 | 46.98 / 0.9889 | 46.10 / 0.9871 |
> | Ours | $\bf{47. 64}$ / $\bf{0 .9891}$ | $\bf{47.10}$ / $\bf{0.9890}$ | $\bf{46 .33}$ / $\bf{0. 9877}$ |
>
> **W4:The paper does not provide ablation experiments.**
>
> Thank you for the suggestion. We evaluate the different number of paths in the network structure of Dy-DCA. We compare the PSNR and the latency tested on mobile GPU for different dynamic paths of Dy-DCA. As shown in the following table, more paths bring higher PSNR by introducing fine-grained data patch groups with more trainable parameters, which sacrifices some latency at the user end. Due to limited time, the results in our paper mainly conducted on a two-path scenario, but we will add the ablation study in the final revision.
>
> | Method | Path number | PSNR | &ensp;Ours(ms)|
> | :--- | :---: | :---: | :-------: |
> |  | | | Min &ensp;  Max |
> | Dy-DCA(WDSR) | 2 | 45.86 | 28  &ensp; 32 |
> | Dy-DCA(WDSR) | 3 | 46.33 | 33 &ensp; 37 |
> | Dy-DCA(WDSR) | 4 | 46.51 | 37 &ensp; 42 |
>
> We also test different video length to show our design is capable of handling longer and more complex video contents. We select 2-min game video in VSD4K [R1] and combine game-45s video and the vlog-45s video together into a 90s-long video (combine-90s). We conduct experiments using WDSR as a backbone with a scaling factor of 4. These two videos are longer and more complex when compared with the videos we show in Table 2 in our paper. As we can see in the following table, the 2-min game video is still maintaining an acceptable PSNR, and the PSNR of combine-90s is close to the average value of overfitting two videos (game-45s and vlog-45s). The results show that our proposed framework can work well even in longer videos with complex contents. We will include the ablation study on different video length in the final revision.
>
> | video | game-45s | vlog-45s | game-2min | combine-90s |
> | :---: | :---: | :---: | :---: | :---: |
> | **PSNR** | 45.86 | 49.84 | 44.72 | 47.34 |
>
> [R1]Liu J, Lu M, Chen K, et al. Overfitting the data: Compact neural video delivery via content-aware feature modulation, ICCV 2021.

---

> ### Author Response · Authors · 2023-11-20
>
> Dear Reviewer nwgm,
>
> Your time and effort in reviewing our submission are greatly appreciated. As the deadline for the ICLR rebuttal approaches on the 22nd, your timely feedback would greatly assist us in preparing a comprehensive response, ensuring that we can thoroughly address any points you have highlighted. Thanks.

---

### Official Review · Reviewer_wfAU · 2023-11-06

**Soundness:** 3 good
**Presentation:** 2 fair
**Contribution:** 3 good
**Rating:** 5
**Confidence:** 3

**Summary:**

The paper proposes dynamic processing of different regions in video frames. Furthermore, the authors propose a compiler that manages the dynamism of tensor shapes.

**Strengths:**

Real-time video super resolution on a mobile device at 1080p is an impressive result.

**Weaknesses:**

Rewriting the paper to be more accessible to non-expert readers would be beneficial. As it stands, the flow of information assumes the reader is familiar with compilers, the current state of SR, and the continuous model adaptation framework, which may not always be the case.

**Questions:**

1. The paper rightly points out that switching the Super-Resolution (SR) model from one chunk to another can be costly. Could the solutions proposing sparse incremental model changes address this issue (refer to [1] for an example)?

2. The unique contributions in Section 2.3 are difficult to comprehend in their current form. How does this categorization of tensors encompass all use cases? What are the limitations of the proposed compiler? Is it applicable solely to SR or also to convolutional models?


[1] Khani, Mehrdad, Vibhaalakshmi Sivaraman, and Mohammad Alizadeh. "Efficient video compression via content-adaptive super-resolution." In Proceedings of the IEEE/CVF International Conference on Computer Vision, pp. 4521-4530. 2021.

---

> ### Author Response · Authors · 2023-11-18
>
> Dear Reviewer wfAU,
>
> Thank you for your review and suggestions on our paper.
>
> **The paper rightly points out that switching the Super-Resolution (SR) model from one chunk to another can be costly. Could the solutions proposing sparse incremental model changes address this issue (refer to [1] for an example)?**
>
> The method proposed by [R1] can adaptively fit different content with incremental model changes in a single model. However, when the video content becomes complicated, the super-resolution quality of [R1] drops dramatically. For example, in the following table, the inter-45s and game-45s have richer information. Compared with our method, the performance of SRVC drops a lot when using WDSR as the backbone with a scaling factor of $\times2$. Especially for gmae-45s, a 31.03 dB is not acceptable for a high-quality video requirement.
>
> Furthermore, our method is based on algorithm and compiler co-design, which is more favorable to practical usage. The operators and data flow can be fully supported by mobile devices.
>
> | **Method** | **vlog-45s** | **inter-45s**| **game-45s** | **Average** |
> |----------------|----------------|----------------|-------------------|------------------|
> | SRVC           | 48.71          | 39.26          | 31.03             | 39.68            |
> | Dy-DCA (Ours)  | 49.84          | 44.76          | 45.86             | 46.82            |
>
> [R1] Khani, Mehrdad, Vibhaalakshmi Sivaraman, and Mohammad Alizadeh. "Efficient video compression via content-adaptive super-resolution."
>
> **The unique contributions in Section 2.3 are difficult to comprehend in their current form. How does this categorization of tensors encompass all use cases? What are the limitations of the proposed compiler? Is it applicable solely to SR or also to convolutional models?**
>
> We classify the operators used for modern DNNs (specifically 150 operators that are included in the widely-used ONNX (Open Neural Network Exchange) format) into 4 categories. Therefore, we do consider all use cases.
>
> Essentially, our proposed algorithm and compiler co-design framework can target all dynamic situations. It is a general optimization framework for dynamic neural networks. Besides the SR task, we further test it on object detection tasks with YOLO-V6 [R1] and language model CodeBERT [R2] with dynamic input size. We also compare our method with MNN, and the results show that our approach works well on other types of network architectures (e.g., CNN and transformer) and tasks, and consistently outperforms MNN in terms of storage and speed.
>
> |    | MNN(MB)  | &ensp; &ensp;Ours(MB) | MNN (ms) | &ensp; &ensp; Ours (ms) |
> | :-: | :-----: | :-----: | :-----: | :-----: |
> | **Model** |   Min &ensp; Max |  Min &ensp; Max | Min &ensp; Max | Min &ensp; Max |
> | YOLO-V6 | 148  &ensp; 404 | 89  &ensp;  206 | 168 &ensp;  925 | 118  &ensp;  546 |
> | CodeBERT | 25  &ensp;  54  | 21  &ensp;  41| 125  &ensp; 1265  | 102  &ensp;  452 |
> | Our framework speedup |  | $1.19 \times-1.96 \times$ | | $1.23 \times-2.80 \times$ |
>
> [R1] Li C, Li L, Jiang H, et al. YOLOv6: A single-stage object detection framework for industrial applications
>
> [R2] Feng Z, Guo D, Tang D, et al. Codebert: A pre-trained model for programming and natural languages

---

> ### Author Response · Authors · 2023-11-20
>
> Dear Reviewer wfAU,
>
> We appreciate the time and effort you have dedicated to reviewing our submission. As the deadline for the ICLR rebuttal approaches on the 22nd, we kindly request your feedback at your earliest convenience. This will allow us sufficient time to further address any potential questions or concerns you may have raised. Thanks.

---

### Official Review · Reviewer_J3a9 · 2023-11-11

**Soundness:** 2 fair
**Presentation:** 2 fair
**Contribution:** 3 good
**Rating:** 5
**Confidence:** 2

**Summary:**

Deep neural networks (DNNs) are increasingly utilized for video resolution upscaling in modern video distribution systems, enhancing quality and efficiency by overfitting each video chunk with super-resolution (SR) models. However, the high number of models and chunks required causes substantial overhead in model switching and memory usage. To tackle this, a new method, Dy-DCA, employs a single dynamic deep neural network with a content-aware pipeline, significantly reducing the model count. This approach not only improves performance and quality (measured by PSNR) on standard mobile devices but also achieves a 1.7× speedup and up to 1.61× memory savings.

**Strengths:**

* Reducing one dynamic DNN to minimizing the switching overhead is clever.
* Significant improvement in FPS on off-the-shelf mobile phones while maintaining PSNR quality.

**Weaknesses:**

* The presentation of the paper, particularly in Section 2.2 and Section 2.3, needs significant improvement due to a lack of details.

**Questions:**

This is more of a general comment than a specific question. The on-device super-resolution application presented in this work is intriguing, as it enhances video quality using a single dynamic DNN and achieves high FPS through model-compiler co-design. While I understand the constraints of the page limit, the current description of the proposed system (incl. Sections 2.3.3 and 2.3.4) is overly terse and challenging to comprehend for a non-expert reviewer. Therefore, a substantial revision is required to improve the presentation. There are also typos, which requires more careful proofreading.

---

> ### Author Response · Authors · 2023-11-18
>
> Dear Reviewer J3a9,
>
> We sincerely appreciate your thoughtful comments about our work.
>
> **The presentation of the paper, particularly in Section 2.2 and Section 2.3, needs significant improvement due to a lack of details.**
>
> Thank you for your suggestion. We understand the importance of providing comprehensive and detailed information in these sections to enhance the clarity and depth of our work. We are committed to undertaking thorough revisions for Sections 2.2 and 2.3.

---

> ### Author Response · Authors · 2023-11-20
>
> Dear Reviewer J3a9,
>
> Thank you very much for taking the time to review our submission. As the deadline for the ICLR rebuttal approaches on the 22nd, we eagerly await your thoughts on our rebuttal and any further considerations you may have. Thanks.